Requirements for data integration platforms in biomedical research networks: a reference model

Ganzinger Matthias matthias.ganzinger@med.uni-heidelberg.de
Knaup Petra
Institute of Medical Biometry and Informatics, Heidelberg University , Heidelberg , Germany
Juan Hsueh-Fen
Electronic publication date: 2015 Feb 5
Publication date: 2015
Volume: 3
Electronic Location ID: e755
Received 2014 Dec 12; Accepted 2015 Jan 19
Copyright: © 2015 Ganzinger and Knaup
Copyright year: 2015
Copyright holder: Ganzinger and Knaup
License: This is an open access article distributed under the terms of the Creative Commons Attribution License, which permits unrestricted use, distribution, reproduction and adaptation in any medium and for any purpose provided that it is properly attributed. For attribution, the original author(s), title, publication source (PeerJ) and either DOI or URL of the article must be cited.
License URL: https://creativecommons.org/licenses/by/4.0/

Keywords: Research network, Reference model, Data integration, Biomedical informatics, Service-oriented architecture

Funding: Deutsche Forschungsgemeinschaft SFB/TRR 77 This work was funded by the SFB/TRR 77 “Liver Cancer. From Molecular Pathogenesis to Targeted Therapies” of the Deutsche Forschungsgemeinschaft (DFG, http://www.dfg.de). The funders had no role in study design, data collection and analysis, decision to publish, or preparation of the manuscript.

==============================
Biomedical research networks need to integrate research data among their members and with external partners. To support such data sharing activities, an adequate information technology infrastructure is necessary. To facilitate the establishment of such an infrastructure, we developed a reference model for the requirements. The reference model consists of five reference goals and 15 reference requirements. Using the Unified Modeling Language, the goals and requirements are set into relation to each other. In addition, all goals and requirements are described textually in tables. This reference model can be used by research networks as a basis for a resource efficient acquisition of their project specific requirements. Furthermore, a concrete instance of the reference model is described for a research network on liver cancer. The reference model is transferred into a requirements model of the specific network. Based on this concrete requirements model, a service-oriented information technology architecture is derived and also described in this paper.

Introduction

Current biomedical research is supported by modern biotechnological methods producing vast amounts of data (Frey, Maojo & Mitchell, 2007; Baker, 2010). In order to get a comprehensive picture of the physiology and pathogenic processes of diseases, many facets of biological mechanisms need to be examined. Contemporary research, e.g., investigating cancer, is a complex endeavor that can be conducted most successfully when researchers of multiple disciplines cooperate and draw conclusions from comprehensive scientific data sets (Welsh, Jirotka & Gavaghan, 2006; Mathew et al., 2007). As a frequent measure to support cooperation, research networks sharing common resources are established.

To generate added value from such a network, all available scientific and clinical data should be combined to facilitate a new, comprehensive perspective. This requires provision of adequate information technology (IT) which is a challenge on all levels of biomedical research. For example, it is inevitable for research networks to use an IT infrastructure for sharing data and findings in order to leverage joint analyses. Data generated by biotechnological devices can only be evaluated thoroughly by applying biostatistical methods with IT tools.

However, data structures are often heterogeneous, resulting in the need for a data integration process. This process involves the harmonization of data structures by defining appropriate metadata (Cimino, 1998). Depending on the specific needs and data structures of the research network, often a non-standard IT platform needs to be developed to meet the specific requirements. An important requirement might be the protection of data in terms of security and privacy, especially when patient data are involved.

In the German research network SFB/TRR77—“Liver Cancer. From Molecular Pathogenesis to Targeted Therapies” it was our task to explore the most appropriate IT-architecture for supporting networked research (Woll, Manns & Schirmacher, 2013). The research network consists of 22 projects sharing common resources and research data. To provide this network with a data integration platform we implemented a service-oriented architecture (SOA) (Taylor et al., 2004; Papazoglou et al., 2008; Wei & Blake, 2010; Bosin, Dessì & Pes, 2011). The IT system is based on the cancer Common Ontologic Representation Environment Software Development Kit (caCORE SDK) components of the cancer Biomedical Informatics Grid (caBIG) (Komatsoulis et al., 2008; Kunz, Lin & Frey, 2009). The resulting system is called pelican (platform enabling liver cancer networked research) (Ganzinger et al., 2011). Transfer of these data sharing concepts to other networks investigating different disease areas is possible.

We consider our research network as a typical example for a whole class of biomedical research networks. To support this kind of projects, we provide a framework for the development of data integration platforms for such projects. Specifically, we strive for the following two objectives:

Objective 1: Provide a reference model of requirements of biomedical research networks regarding an IT platform for sharing and analyzing data.

Objective 2: Design a SOA of an IT platform for our research network on liver cancer. It should implement the reference model for requirements. While this SOA is specific to this project, parts can be reused for similar projects.

Methods

For the design of a data integration platform it is important to first assess the requirements of the system’s intended users. To support this task, we developed a reference model for requirements. A reference model is a generic model which is valid not only for a specific research network but for a class of such organizations. For the development of the reference model, we used the research network on liver cancer as primary source. These requirements were consolidated and abstracted to get a generic model that can be applied to other research networks.

In the first step, a general understanding of the network’s aims and tasks was acquired by analyzing written descriptions of the participating projects. In addition, questionnaires were sent to the principal investigators to capture the data types and data formats used within the projects. In the second step, projects were visited and their research subjects and processes were captured by interviewing project members.

For the reference model, we use the term goal to describe the highest level of requirements. This is in accordance with ISO/IEC/IEEE 24765, where goal is defined as “an intended outcome” (ISO/IEC/IEEE, 2011). In contrast, requirement is defined as “a condition or capability needed by a user to solve a problem or achieve an objective” (ISO/IEC/IEEE, 2011). In our reference model, each requirement was related to a goal, either directly or indirectly. Requirements were then mapped to concrete functions in the resulting data integration system. On the other hand, goals were used to structure requirements and usually do not lead to a specific function of the system.

To provide a more detailed characterization of the goals, we provide a standardized table for each of them which covers the reference number, name, description and weight of the goal. Table 1 shows the structure of such a table. The complete set of tables for all the goals is available in Table S1.

Table 1 Schema for documenting reference requirements and goals.

Feature	Explanation	
Number	Number for uniquely identifying requirements	
Name	Name of the requirement	
Description	Verbal description of the requirement and its properties	
Weighting	Importance of the requirement for fulfilling the goals of the project (low, medium, high)	

The requirements are documented in the same way as the goals. Figure 1 shows a Unified Modeling Language (UML) diagram with all elements used for describing of both goals and their subordinated requirements (Object Management Group, 2012). As for the goals, we provide a set of tables with more detailed descriptions for all requirements in Table S1 In total, we identified 15 requirements for the reference model.

Figure 1 Overview of the UML elements used in requirements diagrams.

The instantiation of the reference model for requirements to meet the needs of a specific research network provides the basis for the architecture of the desired data integration and its subsequent implementation. We provide a concrete instance of a reference model as well as the resulting IT-architecture in this manuscript.

For the research described in this paper, ethics approval was not deemed necessary. This work involved no human subjects in the sense of medical research, as e.g., covered by the Declaration of Helsinki (World Medical Association, 2013). At no time were patients included for survey or interview. Data was only acquired from scientists regarding their work and data, but no personal or patient related data were gathered. Participants were not required to participate in this study; they consented by returning the questionnaire. No research was conducted outside Germany, the authors’ country of residence. However, in other countries the approval of an institutional review board or other authority might be necessary to apply the reference model.

Results

In this section we describe the reference model for requirements. Then, we show how a concrete model for requirements and an IT-architecture is derived from this reference model. The reference model for requirements is an abstract model and thus a universally usable artefact. It is mapped in several steps to the network specific system architecture.

Reference model

The reference model for requirements covers five reference goals (RG). An overview of the goals and their relations is shown in Fig. 2 by means of a UML requirements diagram. The reference goals are:

• Conduct research project (reference goal RG1): The ultimate goal of a research network is to fulfill the intended research tasks. This usually corresponds to the project specification of the funding organization.

• Answer research questions (reference goal RG2): Each research network has specific research questions it pursues to answer. These questions frame the core of the network and led to its establishment in the first place.

• Create, store, and retrieve data (reference goal RG3): Research networks need data to conduct the project. Thus, it is necessary to generate and handle them.

• Analyze data (reference goal RG4): To generate knowledge out of the data it is necessary to analyze them.

• Control data access and usage (reference goal RG5): Research networks need to protect their data. This includes the prevention of unauthorized access to protected data like patient data as well as aspects of intellectual property rights that need to be respected by authorized users as well.

These goals are ordered in a hierarchical structure: Goal RG1 acts as the root node, which has the two sub goals RG2 and RG3. Goal RG4 is subordinated to Goal RG2, whereas Goal RG5 is a sub goal of Goal RG3.

Figure 2 Reference model for goals of a research network.

Each goal has several requirements. In total, the reference model contains 15 reference requirements (RR). A UML diagram with all reference goals and reference requirements is shown in Fig. 3. Reference requirements are associated with the reference goals as follows:

Figure 3 Reference model for goals and requirements of a research network.

Goal 3 is associated with the reference requirements to create data (RR1) and to retrieve external data (RR2). These two requirements respect possible sources of data necessary for the research network. Reference requirement RR3, represent data, is further defined by its subordinate requirements define syntax (RR4), define data model (RR5), identify data (RR6), and define semantics (RR7).

Goal 5, control data access and usage, has two aspects, which are represented by reference requirements RR8 and RR9. RR8 requires the creator’s contribution in the generation of data for the research network to be recognized when data are used by others. As a consequence, even users with legitimate access to the system have to adhere to usage regulations. These regulations should be checked and enforced by the system as far as possible. In contrast, RR9 covers the requirement to protect data from unauthorized access.

The second group of reference requirements covers data analysis. At the highest level we identify goal 2, answer research questions. It is associated downstream with goal RG4, analyze data. Goal RG4 is composed of two reference requirements: RR11 integrate data and RR13 define analytical process. RR11 is associated with RR3, since the technical provision of data within the research network is of great relevance for the integration of data. RR13 has subordinate requirement RR12, define analytical methods, which covers the low-level data analysis methods. RR14 and RR15 cover two distinct instances of RR13: RR14 describes static workflows with all process steps being fixed. In this case, the order of analytical steps and data sources used cannot be changed by the users of the system. In contrast, RR15 considers dynamic workflows, allowing users to compose analytical steps and data sources as needed. Since the type of data involved in a dynamic workflow is not known upfront, this reference requirement is more demanding in terms of the semantic description of data sources. Precise annotation of data sources is necessary in order to perform automated transformations for matching different data fields.

RR13 is further associated with RR10, show results. RR10 covers the requirement to present the results of the analysis adequately. Thus, it partially fulfils goal 2, answer research questions.

The reference model for requirements is the basis for a network specific requirements model. We present an example for creating such a model and all the following steps in the next section. All goals and requirements from the reference model are mapped to network specific instances. In this process, elements of the reference model are checked for their applicability to the specific research network. Further special requirements of the network are considered at this point as well.

The network specific requirements model is then mapped to system properties. These are qualities contributed to the system by different components. At first, we consider abstract components instead of specific products. For example, in a research network, reference requirement RR1 create data might be mapped to a system property automated creation of data services. This property is then mapped to the specific component responsible for the implementation of this property.

In a second step, the abstract components are mapped to specific components in accordance with the research network’s requirements. Specific components can be preexisting modules with a product character, software development frameworks providing specific functionality, or newly developed components.

In a final modelling step a distribution model of the components is created. All components need to be mapped to system resources down to the hardware level. Among others, the following aspects have to be considered is this step:

• Security: Components with high security requirements should be isolated against other, less sensitive components and thus be run on a separate system node.

• Performance: All components must be distributed in a way that availability of sufficient system resources is ensured.

• Maintainability: To ensure that the possibly complex distributed system can be managed efficiently, components should be grouped together in a sensible way.

Sample application: pelican

We now describe a sample application of our reference model within the research network SFB/TRR77 on liver cancer. Further we describe two specifications for metadata we developed for the research network.

Specific model for requirements

In this section we summarize key requirements specific to our research network. The complete list of requirements is shown in Table S2.

The first goal of the research network, an instance of reference goal RG1, is defined by its research assignment of gaining a deeper understanding of the molecular basis of liver cancer development. This spans research on the chronic liver disease to progression of metastatic cancer. Further, the research network aims to identify novel preventive, diagnostic and therapeutic approaches on liver cancer. Subordinated to goal G1 is G3, the instance of reference goal RG3 regarding the data necessary for the network. Since molecular processes play a major role within the research network, genomic microarray data are of central importance. They are complemented by imaging data like tissue microarray (TMA) data and clinical data.

Goal G2, answering research questions, is characterized by the following two questions:

• Which generic or specific mechanisms of chronic liver diseases, especially of chronic virus infections and inflammation mediated processes predispose or initiate liver cancer?

• Which molecular key events promoting or keeping up liver cancer could act as tumor markers or are promising targets for future therapeutic interventions?

Goal G5 requires making the data available for cross project analysis within the network, but to protect data against unauthorized access at the same time. Especially important to the members of the project is the requirement R8, subordinated to goal G5: The projects contributing data to the network require keeping control over the data in order to ensure proper crediting of their intellectual property. Thus, they require fine-grained rules for data access control. Depending on the type of data, they should be available only to specific members of the network, to all members of the project or the general public.

System architecture

To acknowledge the project’s requirement R8, to keep the ownership over their data, a federation was used as the underlying concept of the system architecture. Technically, pelican implements a SOA. All data sources of the projects are transformed into data services and made available to the research network. The data services stay under the control of the contributing project. This can even go as far as running the service on computer hardware on the projects’ premises. Data services are complemented by analytical services. All services are described by standardized metadata to help finding appropriate services and allow for automated access to the services’ interfaces. Using a web-based user interface, researchers can chain data services and analytical services to answer specific research questions.

Component model

The requirements are mapped to system properties first. In the next step, components are identified to provide these properties as module of the new system. In Table 2 we show the complete chain of mappings from requirements over system features to components. Each component is realized either by a readily available product or by a newly developed module. In Table 3 we give an overview of our components.

Table 2 Mapping of requirements to corresponding system features and components.

Requirement	System feature	Component	
Requirements associated with G2	
R1 Create data	Automated creation of data services	Data service framework	
R2 Retrieve external data	Integration of external data services	Data service framework, portal	
R3 Represent data	Data service, document service	Data service framework, documentmanagement system	
R4 Define syntax	Service description	Data service framework	
R5 Define data model	Defined data model	Data service framework	
R6 Identify data	Provisioning of information on service location	Meta data directory	
R7 Define semantics	Definition of controlled vocabulary and ontologies	Terminology server	
Requirements associated with G4	
R8 Administrate intellectual property	Log data usage	Portal, data service framework	
R9 Protect data	User authenticationUser authorization	Portal, security service Portal, data service framework, security service	
Requirements associated with G5	
R10 Show results	Data specific portlets	Portal	
R11 Integrate data	Analytical services	Portal, statistics service, data service framework	
R12 Define analytical methods	Statistical methods	Statistics service	
R13 Define analytical process	Documentation service	Document management system	
R14 Static workflow	Workflow in portal application	Portal	
R15 Dynamic workflow	Flexible pipeline	Pipeline management	

Table 3 Specification of concrete implementation components for the elements of the component model.

Abstract component	Implementing component	
Portal	Liferay	
Data service framework	caCORE SDK	
Meta data directory	Internal development (based on caCORE SDK)	
Terminology server	TemaTres	
Security service	caCORE SDK, LDAP	
Statistics service	R	
Document management system	Alfresco	
Pipeline management	Galaxy (planned)	

The portal component provides the user interface to the system. It is implemented using the open source software Liferay (http://www.liferay.com, accessed: 2014-07-03) (Sezov, Jr, 2012). Liferay provides a number of functions affecting several components of our model. Thus, we provide a decomposition of the portal components in Fig. 4. One important subcomponent of the portal is the document management system. It is realized by the Alfresco component (http://www.alfresco.com, accessed: 2014-07-03) (Berman, Barnett & Mooney, 2012). The user interface of Alfresco can be integrated into the Liferay portal or be accessed with a separate unified resource locator (URL). The portal provides user management functionality to control access to portal pages and components like portlets (Java Community Process, 2008). However, the user account information including username, passwords, and others is stored in a separate component using the Lightweight Directory Access Protocol (LDAP). Thus, it is possible for all components of the SOA-network to commonly access the users’ identity information.

Figure 4 Structure of the component portal.

Data services are generated by using caCORE SDK (Wiley & Gagne, 2012). With caCORE SDK it is not necessary to program the software for the service in a traditional way. Instead, a UML data model in Extensible Markup Language Metadata Interchange (XMI) notation has to be prepared (Object Management Group, 2002; Bray et al., 2006). From this model, caCORE SDK generates several artefacts resulting in a deployment packages for Java application servers like apache tomcat (The Apache Software Foundation, 2014). To simplify this process for spreadsheet based microarray data, we developed a software tool to generate the XMI file as well. As a result, a service conforming to the web services specification ready for deployment is generated. For the provision of network specific metadata we chose TemaTres to serve our controlled vocabulary in standard formats like SKOS or Dublin Core (Weibel, 1997; Miles & Bechhofer, 2009; Gonzales-Aguilar, Ramírez-Posada & Ferreyra, 2012). Our analytical services are backed by the open source language and environment for statistical computing called R (R Core Team, 2014). R is integrated into the services using the Rserve component (Urbanek, 2003).

Deployment model

In a final modelling step the components are distributed to the physical resources available for the system. In our case we used two servers with a common virtualization layer based on VMware VSphere server. Thus, all nodes in our deployment model represent virtual machines (VM). Using virtual switches, routers, and firewall appliances we were able to implement our Internet Protocol (IP) network infrastructure. To enhance security, we implement a network zoning model comprised of an internet zone, a demilitarized zone (DMZ) and an internal zone. Figure 5 shows the deployment model in UML notation. The services shown in the model (service 1 to service n) are to be considered as examples, since the concrete number of services is permanently changing. The deployment model also reflects the different levels of control that can be executed by the owners of the data. They range from shared nodes on the common servers over a dedicated VM to deployment on external hardware controlled by the respective projects.

Figure 5 Deployment diagram of the components of the architecture in UML notation.

Discussion

In this manuscript, we describe a reference model for the requirements of research networks towards an IT platform. For many funding programs, including research grants of the European Commission, the collaboration of several research organizations at different sites is mandatory. This leads to a structural similarity to our research network on liver cancer. Even though other research networks will have different research aims, there are still requirements that are common to most networks. Since the reference model already covers a basic set of requirements, it allows future research networks to focus on defining specific requirements distinguishing them from other networks.

Users of the reference model are responsible for assessing the reference model’s applicability to their project-specific needs. The reference model is based on data of a real research network that were generalized. To avoid bias in the model that might hinder transferability, we incorporated different views in the process of constructing the model. However, the transferability of the model to another context is, as for any model, limited. As a consequence, future research networks will have to derive a project specific instance of the reference model to reflect the corresponding characteristics of the project. The reference model is a tool intended to help its users to create a concrete model covering the requirements of a research network with a high degree of completeness. The reference model provides guidance for this task. We expect that it will help in reducing the effort to acquire all requirements.

A common technique used in requirements engineering for software is use case modelling (Jacobson, 1993; Bittner & Spence, 2003). A use case is a compact scenario describing certain aspects of how a technical system behaves and how it interacts with other actors like its users. A use case model is a way of capturing requirements in an interactive way. Thus, a use case model can be developed further into a requirements model. In our context we found it hard to apply the use case approach since the researchers in our network mostly have biomedical backgrounds and thus are not familiar with software development. Nevertheless, use case modelling might be a helpful tool for other research networks when applying our reference model.

We applied the reference model successfully to a research network on liver cancer. Some specific requirements in this network led to the decision to set up a federated system allowing for a maximum of control of the individual projects over their respective data. The system was implemented as a service-oriented architecture using, among others, components of the caBIG project. A public version of our system with limited functionality is available at https://livercancer.imbi.uni-heidelberg.de/data. There, a gene symbol like BRCA1 can be entered into the search field and services are invoked for data retrieval as discussed in this paper. Other projects can benefit from this architecture as well, but the architecture is tailored to research networks with the requirements of federating data as data services. With this architecture, we try to acknowledge the data protection requirements of the participating projects. Still, further research regarding the use of data and crediting creatorship of data is necessary. First steps were made as part of this project (He et al., 2013).

In case the requirements regarding data control are more relaxed, an alternative would be to keep the data in a central data warehouse instead of the federation. In that case, i2b2 might be a suitable component to provide the data warehouse component (Murphy et al., 2007). Such a centralized approach also affects how and when data are harmonized: In a central research data warehouse data are harmonized at the time of loading the database, which ideally leads to a completely and consistently harmonized database. In a service-oriented approach data are provided by means of data services as they are. All services are described by corresponding metadata, enabling automated transformation of the data at time of access.

Our sample research network concentrates more on basic research than clinical application. In the future, we plan to apply our reference model to further projects with a stronger translational component. By doing so, we will be able to reevaluate the framework in a more clinical context.

Supplemental Information

Table S1 Reference goals and requirements

Click here for additional data file.

Table S2 Goals and requirements for pelican

Click here for additional data file.

Additional Information and Declarations

Competing Interests

Author Contributions

Human Ethics

The authors declare there are no competing interests.

Matthias Ganzinger conceived and designed the experiments, performed the experiments, analyzed the data, wrote the paper, prepared figures and/or tables, reviewed drafts of the paper.

Petra Knaup wrote the paper, reviewed drafts of the paper.

The following information was supplied relating to ethical approvals (i.e., approving body and any reference numbers):

For the research described in this paper, ethics approval was not deemed necessary. This work involved no human subjects in the sense of medical research, as e.g., covered by the Declaration of Helsinki (World Medical Association, 2013). At no time patients were included for survey or interview. Data was only acquired from scientists regarding their work and data, but no personal or patient related data were gathered. Participants were not required to participate in this study; they consented by returning the questionnaire. No research was conducted outside Germany, the authors’ country of residence. However, in other countries the approval of an institutional review board or other authority might be necessary to apply the reference model.

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
