# Peer review of "Requirements for data integration platforms in biomedical research networks: a reference model"

_PeerJ, doi:10.7717/peerj.755_

## Round 0.1 · original submission · Minor Revisions

This manuscript will be accepted if you address all the reviewers' comments. When you submit your revised manuscript, please also provide an executable IT platform for the reviewers to test.

·

Basic reporting

No Comments

Experimental design

No Comments

Validity of the findings

No Comments

Additional comments

This paper proposed a reference model for requirements. Based on this reference model, specific requirements for developing a system for cross-projects collaboration within biomedical research networks can be captured efficiently. As a demonstration of usage, the implementation of this reference model for requirements to a research network on liver cancer was then illustrated in this study.
Requirements analysis is critical in developing a system. The proposed reference model for requirements may help provide a guidance for developing a system for research networks. So I would like to recommend this manuscript accepted with some revisions listed below.
1. There are useful tools for requirements analysis, for example, the use case analysis. What is the relation between requirement diagrams and use cases and how does the proposed reference model for requirements accomplish use cases or complement use cases? Is the proposed modeling steps sufficient for developing a data integration platform?
2. Following are several typos or inconsistencies need to be checked by the authors:
1) Line 137, it is Goal 3 that was described in the paragraph, and misplaced by Goal 2. Seemly, Line 143 in the next paragraph, I think it would be RR8 instead of RR9 regarding usage regulations.
2) In Figure 3, words below RG3 should be adjusted to make it readable, and typo should be checked, e.g. “intellectual” of RR8.
3) In supplementing information S2, since all reference requirements in the proposed reference model was mapped to requirements of the specific research network, the prefixes of Number RR14 and RR15 should be “R” instead.
4) Line 199, it is RG3 instead of R3.
5) In Table 2, “data service” in the column Component for Requirements R8 and R9 would be better replaced by “data service framework” for consistency if they are the same components.
6) Figure 5 is a little bit confusing because of the two Service provider 1 in the Internal Zone in the diagram. Are they from the same service provider? Or is there another service provider?
7) Line 262, I would suggest adding abbreviation of demilitarized zone (DMZ) in the sentence so that it can be traced back while reading DMZ in Figure 5.

Reviewer 2 ·

Basic reporting

No Comments

Experimental design

No Comments

Validity of the findings

No Comments

Additional comments

The authors proposed a data integration platform in biomeical research networks. The concept is well presented. However, it would be better if the authors can provide an executable IT platform for the reviewer to test. In that case, the usefulness of the platform can be judged in a rigorous way.

---

## Round 0.2 · accepted · Accept

The authors have addressed all the reviewers' comments, so it can be accepted now.

·

Basic reporting

No Comments

Experimental design

No Comments

Validity of the findings

No Comments

Additional comments

It seems that the revisions authors made for Table 2 did not appear successfully.

Reviewer 2 ·

Basic reporting

No Comments

Experimental design

No Comments

Validity of the findings

No Comments